# Visualization of Slag Data for Efficient Monitoring and Improvement of Steelmaking Slag Operation in Electric Arc Furnaces, with a Focus on MgO Saturation

**Marcus Kirschen**

Thermal Process Engineering, University of Bayreuth, Universitätsstrasse 30, D-94557 Bayreuth, Germany; marcus.kirschen@uni-bayreuth.de

**Abstract:** Frequent slag sampling and analysis is still the most common method used to investigate and improve slag operation in electric arc furnaces (EAFs) for low-alloyed carbon steelmaking. An MgO saturation diagram for EAF slags was derived from phase equilibrium calculations in the system $CaO–MgO–FeO–SiO_2–5\%Al_2O_3$ to provide monitoring and interpretation of the slag data with respect to control of MgO saturation, FeO reduction, dissolution from the MgO-based refractory lining, and unusual losses of repair mixes. Examples from 14 industrial EAFs are given.

**Keywords:** electric arc furnace; process slag; saturation model; process improvement

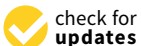



## 1. Introduction

Steelmaking based on the melting of steel scrap and other ferrous raw materials in electric arc furnaces (EAFs) contributed approximately 523 million tons, i.e., 28%, of global steel production in 2019 [1]. The share of EAF steelmaking is growing globally due to increasing steel scrap recycling capacity in developed economies and very flexible production regarding the use of ferrous raw materials or alloys with volatile market demand (examples in Figure 1). Recycling of steel scrap and the use of direct reduced iron (DRI) in EAFs are important options to control the specific $CO_2$ emissions of the global steel industry.

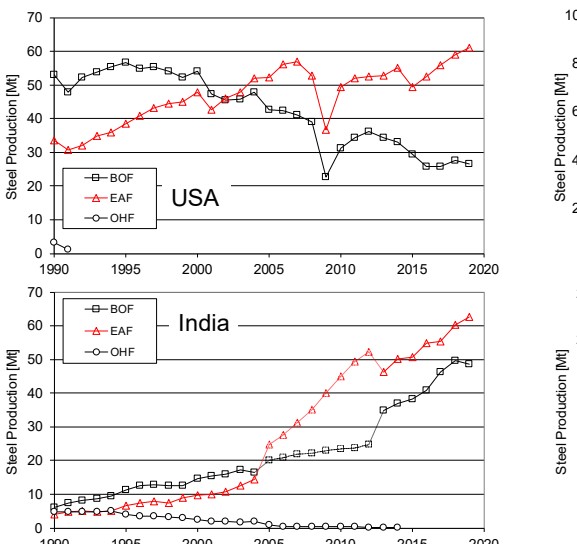

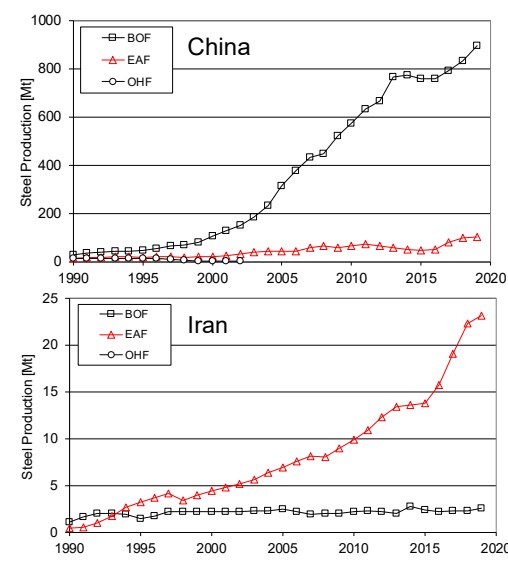

**Figure 1.** Four examples of electric arc furnace (EAF) steel production on different levels in the global steel industry (basic oxygen furnace, BOF; open hearth furnace, OHF); data from World Steel Association [1].

Process control and optimization in EAFs with various designs (e.g., alternating or direct current EAFs, and EAFs with scrap preheating shafts or conveyors for continuous charging of scrap and DRI) requires reliable data from the inside, which are still challenging to obtain. The available data are from regular steel sample analysis and temperature, C content, and oxygen measurements at the final process stage and, commonly, slag samples before tapping. The importance of appropriate slag operation in EAFs was realized early for optimized efficiency of energy transfer from the electric arcs by foaming slag ([2] and references therein). Pretorius and others provided helpful guidelines and models for slag operation in basic oxygen furnaces (BOFs), EAFs, and ladles during steel refinement [3–5]. Frequent slag sampling and analysis has been implemented as an efficient process monitoring tool to ensure efficient operation under foaming slag conditions with an appropriate slag viscosity, requiring control of the slag composition at MgO saturation.

### 1.1. Slag Operation in EAFs

Slag formers such as lime, dolomitic lime, or doloma are charged with steel scrap or alternative ferrous raw materials such as DRI in order to dissolve the oxidation products and residues of the steel scrap, including $SiO_2$, $Al_2O_3$, $Cr_2O_3$, $P_2O_5$ (e.g., [6]), and $TiO_2$. FeO is partially reduced in the slag layer by mixing with carbon-containing steel melt or by injecting coal or carbon fines (e.g., [7]). The efficiency of energy transfer from the electric arcs to the molten metal is increased when submerged in a foaming slag layer with increased volume (e.g., [8,9]) that is stable at a certain range of apparent viscosity, surface tension, and sufficient CO gas generation (e.g., [10–12]). The viscosity of the basic EAF slag is increased to the required range using finely dispersed periclase crystals, i.e., when the slag composition is MgO saturated [4,5]. Consistently, optimum slag foaming is observed at EAF slag compositions with basicity $B_2 = x_{CaO}/x_{SiO2} \approx 2$ [4,5], i.e., heterogeneous slag foaming [9,13], in order to compensate the viscosity decreasing effect of FeO (e.g., [14]). The complex behavior of FeO in slag foaming was clarified by considering the composition dependence of the $Fe^{3+}/Fe^{2+}$ ratio on the surface tension [15]. Other basicity concepts comprise the decreasing influence of $Al_2O_3$ on basic slag properties, e.g., $B_3 = x_{CaO}/(x_{SiO2}+x_{Al2O3})$ or $B_4 = (x_{CaO} + x_{MgO})/(x_{SiO2} + x_{Al2O3})$.

### 1.2. MgO Saturation of EAF Slags—Implications for the Refractory Lining Lifetime

Besides operating the process slag at good foaming slag conditions, MgO saturation helps to minimize corrosion of the MgO-based hearth and MgO–C sidewall lining. At undersaturated slag compositions, i.e., at MgO activity of <1, MgO is dissolved from the MgO-based hearth and hearth repair mixes or from the MgO–C-based sidewall, depending on the process temperature and duration (e.g., [16]), until MgO saturation of the slag is achieved. High-FeO slags provide increased corrosion potential to the MgO lining by the degradation of periclase to Mg–wüstite [17,18] and lowered slag viscosity.

The loss of MgO from the lining by spalling, e.g., due to thermal shocks, or MgO loss from repair layers results in increased MgO content of the slag, either dissolved at MgO-undersaturated compositions or as dispersed solid periclase at MgO-oversaturated compositions, indicating unwanted and unnecessarily high MgO losses.

Depending on the MgO content and the mass ratio of the charged slag formers, the EAF process is started with an MgO-undersaturated slag composition in most cases. The amount of dissolved MgO in the slag from the beginning of heating until tapping is related to the MgO losses from the lining, due to the MgO mass balance [19]. Therefore, it is recommended to apply MgO-containing slag additives or modifiers as early as possible in order to decrease the corrosion potential of the process slag [4,5,19–22]. This keeps the MgO activity in the slag at high levels, ideally close to 1. An increased mass of hot heel is recommended in order to provide a larger slag mass for buffering increased $SiO_2$ input, for example, due to DRI [18]. Models of MgO saturation are crucial to optimize the input of slag formers and slag operation in EAF steelmaking.

*1.3. MgO Saturation Models for Steelmaking Slags*

Some proposed saturation models determine the MgO concentration as a complex function of the slag composition and temperature, based on saturation experiments on various metallurgical process slags, e.g., [4,23–27]. Pretorius and Carlisle [4], Park and Lee [24], and Tayeb et al. [26] provided the MgO saturation as a function of slag basicity for EAF slags or ladle slags. The typical MgO saturation concentrations in EAFs are approximately 9% MgO in EAF slags with basicity $B_2 = 2$ [4].

Another approach is the derivation of saturation diagrams from published phase diagrams in the $CaO–MgO–FeO–SiO_2–Al_2O_3$ system [4,5], or the application of phase equilibrium software [18,19,28–31] such as Factsage [32] to calculate the required MgO saturation diagrams in the $CaO–MgO–FeO–SiO_2–Al_2O_3$ system for ladle and EAF slags. In all cases, a formula for a restricted range of slag compositions or an *x–y* diagram with a constraint slag composition, e.g., for a constant basicity, was chosen to determine the required MgO saturation for the particular process slag composition. An overview on published MgO saturation models in steelmaking is given in Table 1.

**Table 1.** Published models of MgO saturation of steelmaking slags.

| Author(s) | Target Function | Metallurgical Unit | Variables |
|---|---|---|---|
| Schürmann and Kolm (1986) [23] | MgO saturation | BOF, EAF | $CaO$, $SiO_2$, $FeO$, $Al_2O_3$, $MnO$, T |
| Park and Lee (1996) [24] | MgO saturation | Ladle | $B_3$ basicity |
| Pretorius and Carlisle (1999) [4] | MgO (and $C_2S$) saturation | EAF | $FeO$, $B_2$ and $B_3$ basicity |
| Park (2001) [25] | MgO saturation | BOF | $CaO$, $SiO_2$, $FeO$, $Fe_2O_3$ |
| Brüggmann and Pötschke (2011) [28] | MgO saturation | Ladle | $CaO$, $SiO_2$, $Al_2O_3$ |
| Tayeb et al. (2015) [26] | MgO saturation | BOF, EAF | $CaO$, $SiO_2$, $Al_2O_3$, $FeO$ |

All these models are useful to calculating the MgO saturation of particular slag compositions, either for laboratory investigations or for average values of industrial data sets. Slag compositions in industrial EAFs, however, span a wider range in the composition space $CaO–MgO–FeO–SiO_2–Al_2O_3$ due to variations in raw materials and process conditions. Instead of pointwise assessment of average MgO saturation levels, it is of interest to visualize and investigate the slag compositions of large datasets from industrial EAFs with respect to the complex MgO saturation surface in multicomponent space.

**2. Calculated MgO Saturation Surface in the System $CaO–MgO–FeO–SiO_2–5\%Al_2O_3$**

The databases provided with the Factsage software package are today the best available representation of the binary, ternary, and quaternary diagrams of the $CaO–MgO–FeO–MnO–SiO_2–TiO_2–Al_2O_3$ system. Recently, Tayeb et al. [26] verified the results from Factsage calculations with experimental MgO solubility data at typical slag compositions for BOFs and EAFs. Song et al. [18] presented valuable predictions from a recent Factsage database of MgO saturation concentrations in EAF slags for a significantly larger experimental dataset.

The saturation surface of MgO was calculated using the Factsage database in the system $CaO–MgO–FeO–SiO_2–Al_2O_3$ at 1600 °C and constant 5% $Al_2O_3$ for EAF slags (Figure 2), thus neglecting the minor $TiO_2$ and $Cr_2O_3$ slag components in carbon steelmaking slags. MnO [33] was also neglected here, suggesting the addition of a low amount of MnO to FeO when required for higher precision. The pseudo-ternary diagram in Figure 2 represents a slide section in the five-component space $CaO–MgO–FeO–SiO_2–Al_2O_3$ at 30% CaO and 5% $Al_2O_3$. The chosen section is best suited to investigating the MgO saturation status of EAF slags for low-alloyed steel grades because (1) the MgO saturation field is represented to a maximum extent and (2) the calculated saturation lines are derived at compositions close to the slag data, in contrast to other MgO saturation figures (e.g., [4,5]).

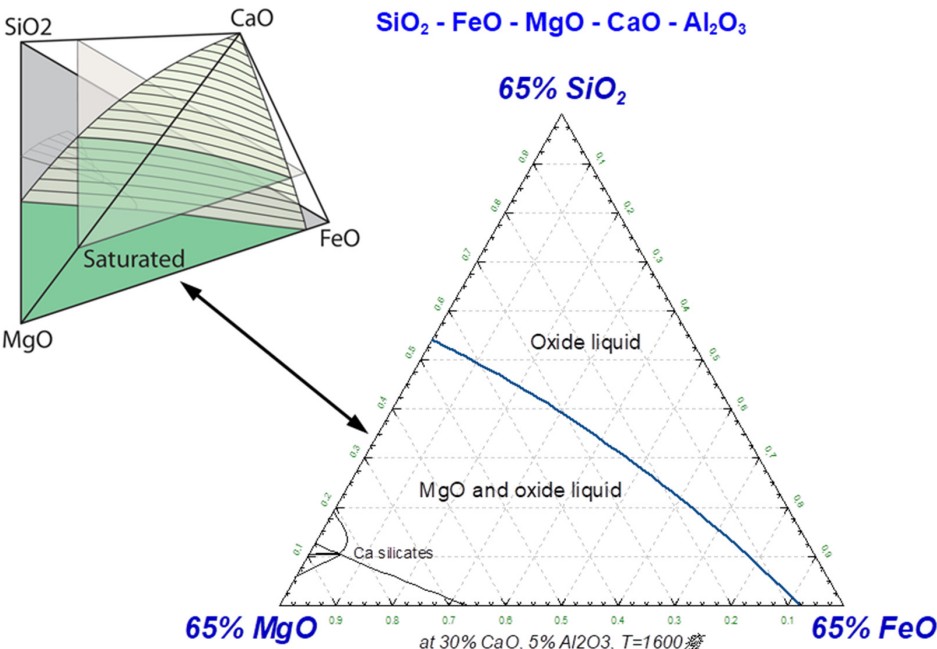

**Figure 2.** The calculated stability field of MgO periclase and Ca silicates on the pseudo-triangular plane at 30% CaO and 1600 °C in the system CaO–SiO$_2$–FeO–MgO–5%Al$_2$O$_3$; all calculations from Factsage™.

The most relevant saturation surface of EAF slags is the MgO saturation (Figure 2). Ca silicates are not stable in EAF slag compositions for low-alloyed carbon steelmaking at temperatures between 1550 °C and 1700 °C due to the >20% FeO and 3–10% Al$_2$O$_3$ content (Figure 2).

In Figure 3, the MgO saturation lines are shown on the pseudo-ternary sections at 25%, 30%, 35%, and 40% CaO at T = 1600 °C, corresponding to the typical CaO range in EAF slags.

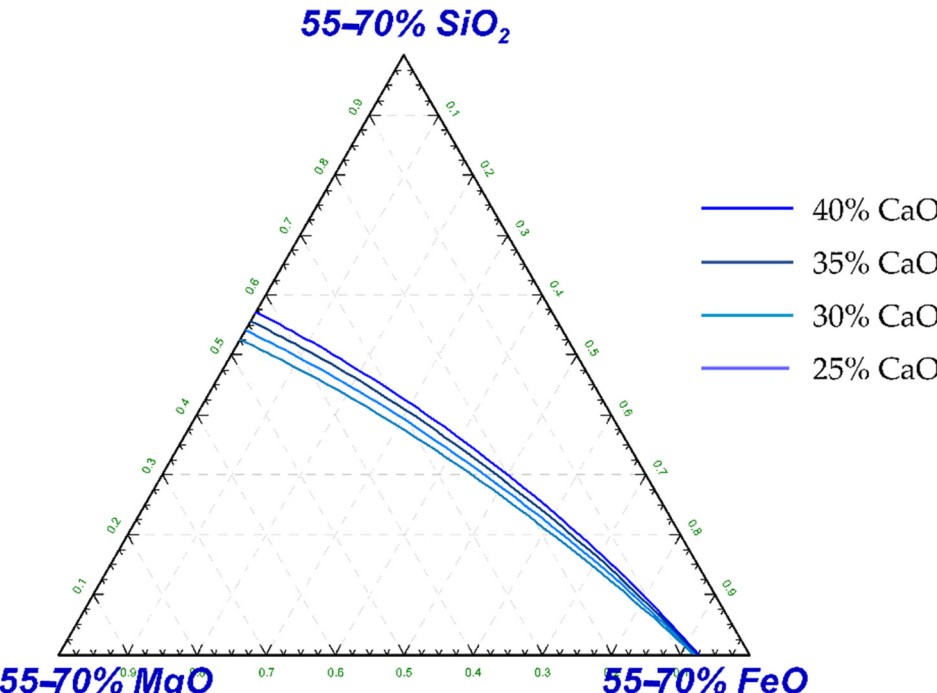

**Figure 3.** Projected MgO saturation lines in the pseudo-ternary sections at 25%, 30%, 35%, and 40% CaO at 1600 °C in the system CaO–SiO$_2$–FeO–MgO–5%Al$_2$O$_3$; all calculations from Factsage™.

### 3. Application to Slag Analysis Data from Carbon Steelmaking EAFs

The average slag compositions of 14 industrial EAFs for low-alloyed carbon steelmaking are given in Table 2. The product portfolio covered rebar and construction steel grades (8 EAFs) to special steel grades (2 EAFs). The applied raw materials were steel scrap (10 EAFs) and blends of steel scrap with DRI (4 EAFs). The sizes of the EAFs ranged from 60 t to >200 t tap weight, located in nine countries worldwide. Slag samples were taken from the EAFs shortly before tapping and analyzed at the steelplant laboratory. Only mislabeled slag data, e.g., those from transport ladles (i.e., FeO < 10% and CaO > 45%) or from raw materials (e.g., lime, DRI), were excluded from the data sets.

**Table 2.** Average slag compositions of 14 industrial EAFs for carbon steelmaking (in wt. %).

| EAF | # Data | CaO | $SiO_2$ | FeO | MgO | $Al_2O_3$ | MnO | $Cr_2O_3$ | $TiO_2$ | Total | Basicity | |
|---|---|---|---|---|---|---|---|---|---|---|---|---|
| 1 [1] | 422 | 26.1 | 16.7 | 29.5 | 10.5 | 8.4 | 5.6 | 1.6 | 0.7 | 99.6 | 1.6 [5] | 1.5 [6] |
| 2 [1] | 359 | 31.1 | 11.6 | 28.1 | 10.6 | 5.4 | 5.0 | 1.1 | 0.6 | 94.4 | 2.7 [5] | 2.5 [6] |
| 3 [1] | 1216 | 25.6 | 13.5 | 34.5 | 11.3 | 6.0 | 6.4 | 2.5 | 0.4 | 100.7 | 1.9 [5] | 1.9 [6] |
| 4 [1] | 471 | 25.6 | 12.1 | 29.7 | 9.4 | 14.5 | 4.6 | 2.1 | n.a. | 97.9 | 2.1 [5] | 1.3 [6] |
| 5 [2] | 149 | 27.3 | 8.8 | 40.2 | 8.3 | 3.5 | 7.0 | 3.2 | 0.4 | 99.4 | 3.1 [5] | 2.9 [6] |
| 6 [3] | 132 | 27.0 | 16.0 | 31.1 | 14.9 | 6.0 | 1.9 | n.a. | 1.2 | 98.0 | 1.7 [5] | 1.9 [6] |
| 7 [1] | 186 | 26.0 | 11.9 | 34.9 | 10.0 | 9.1 | 3.9 | 3.2 | 0.5 | 99.5 | 2.2 [5] | 1.7 [6] |
| 8 [1] | 843 | 35.5 | 17.4 | 16.9 | 14.7 | 7.0 | 6.0 | 1.4 | 0.4 | 98.8 | 2.0 [5] | 2.1 [6] |
| 9 [1] | 296 | 28.8 | 12.2 | 30.7 | 9.6 | 6.1 | 8.0 | 2.9 | 0.5 | 99.3 | 2.4 [5] | 2.0 [6] |
| 10 [1] | 63 | 29.8 | 11.6 | 34.7 | 10.4 | 5.1 | 8.5 | 1.8 | 0.4 | 102.2 | 2.6 [5] | 2.4 [6] |
| 11 [3] | 325 | 28.5 | 19.4 | 33.9 | 9.7 | 3.2 | 0.2 | n.a. | 3.7 | 99.3 | 1.5 [5] | 1.7 [6] |
| 12 [3] | 203 | 36.9 | 17.4 | 30.1 | 7.7 | 5.1 | 0.9 | n.a. | n.a. | 95.5 | 2.1 [5] | 2.0 [6] |
| 13 [4] | 519 | 29.1 | 16.8 | 25.6 | 9.3 | 7.8 | 1.1 | n.a. | n.a. | 89.9 | 1.7 [5] | 1.6 [6] |
| 14 [1] | 219 | 35.5 | 17.5 | 27.6 | 4.1 | 9.3 | 4.2 | n.a. | n.a. | 98.4 | 2.0 [5] | 1.4 [6] |

[1]: rebar and construction steel grades, [2]: specialty steel grades, [3]: 1 with direct reduced iron (DRI); [4]: construction and specialty steel grades, DRI; #: number of slag data; [5]: $B_2 = CaO/SiO_2$; [6]: $B_4 = (CaO + MgO)/(SiO_2 + Al_2O_3)$, n.a.: not available.

Most EAF slags were set to a $B_2$ basicity of 1.9 to 2.4, with only a few exceptions at lower basicity (EAFs 1, 6, 11, and 13) and a few at higher basicity (EAFs 2, 5, and 10). A higher basicity is required, e.g., to keep the P value of the tapped steel below strict limits (e.g., [6,34]).

EAF slag analysis data from various carbon steelmaking EAFs are plotted in the proposed saturation diagram in the system $CaO–MgO–FeO–SiO_2–5\%Al_2O_3$ (Figure 3) in Figures 4–6, without any further data manipulation or data processing of the provided data. The EAF slag data represent multidimensional distributions at or near the MgO saturation.

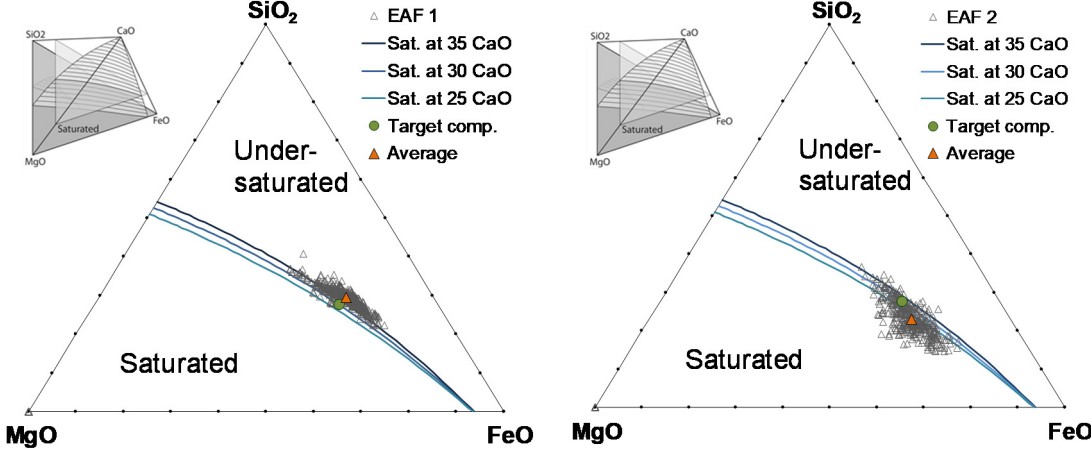

**Figure 4.** Distribution of slag analysis data from EAF heats for carbon steelmaking, in the system $CaO–SiO_2–FeO–MgO–5\%Al_2O_3$; lines indicate MgO saturation at 25% to 35% CaO and 1600 °C.

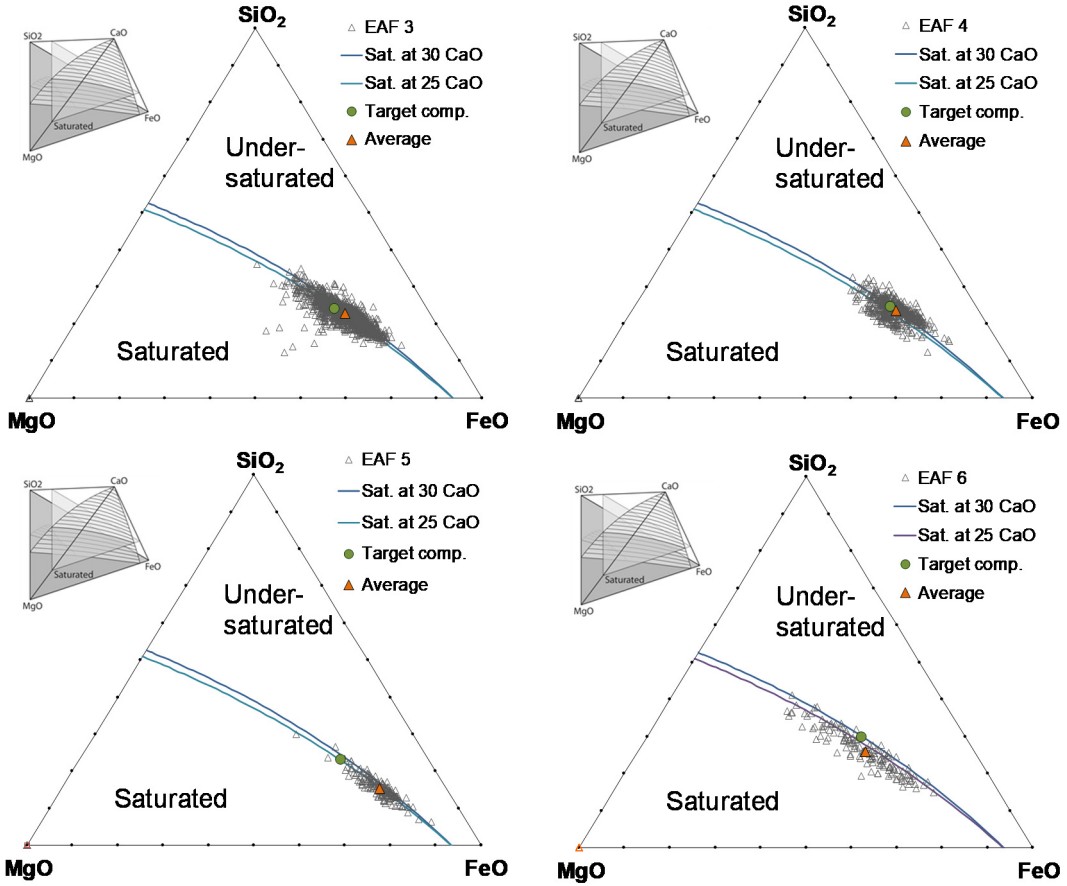

**Figure 5.** Distribution of slag analysis data from EAF heats for carbon steelmaking, in the system CaO–SiO$_2$–FeO–MgO–5%Al$_2$O$_3$; lines indicate MgO saturation at 25% to 35% CaO and 1600 °C.

The slag data of EAF 1 to EAF 6 in Figures 4 and 5 are very close to the MgO saturation surface, which is represented by saturation lines at 25% CaO to 35% CaO, corresponding to the average CaO content of the slag data. In the case where the initial MgO content is lower than the saturation value, some MgO is dissolved from the EAF hearth and sidewall until the MgO saturation surface is reached. Some MgO-oversaturated slags of EAF 3 are most likely related to losses of MgO-based hearth repair mixes or sidewall gunning mixes after a preceding EAF refractory repair. EAF 6 applied mainly doloma as slag former, resulting in slightly MgO-oversaturated slags.

The slag data from EAF 7 to EAF 10 in Figure 6 show a more pronounced scatter in composition. The increased standard deviation of slag compositions indicates lower process control: For example, an increased distribution in FeO content at controlled basicity indicates problems with the carbon–oxygen balance from the injectors and decreased efficiency of FeO reduction by injected carbon fines (e.g., EAF 6 to EAF 8 and EAF 10). EAF 8 was characterized by low slag control in FeO, SiO$_2$, and MgO due to unusually low input of lime/doloma blends, at <25 kg/t. Then, the resulting unusual low slag mass was subject to increased compositional shifts due to usual scrap residues, losses of MgO repair mixes, and overoxidation of the slag (EAF 8).

Some EAF processes were characterized by mainly MgO-undersaturated slag compositions due to the application of lime or dolomitic lime as a slag former, with initial total MgO significantly lower than the saturation concentration, near 9% MgO (Figure 6). Consequently, the slags showed MgO concentrations between the starting MgO content and MgO saturation depending on the FeO content, temperature, and process time (e.g., EAF 11 to EAF 13). Only mechanical losses of MgO repair mixes to the slag produced MgO concentrations that were higher than the MgO saturation (EAF 11 to EAF 13).

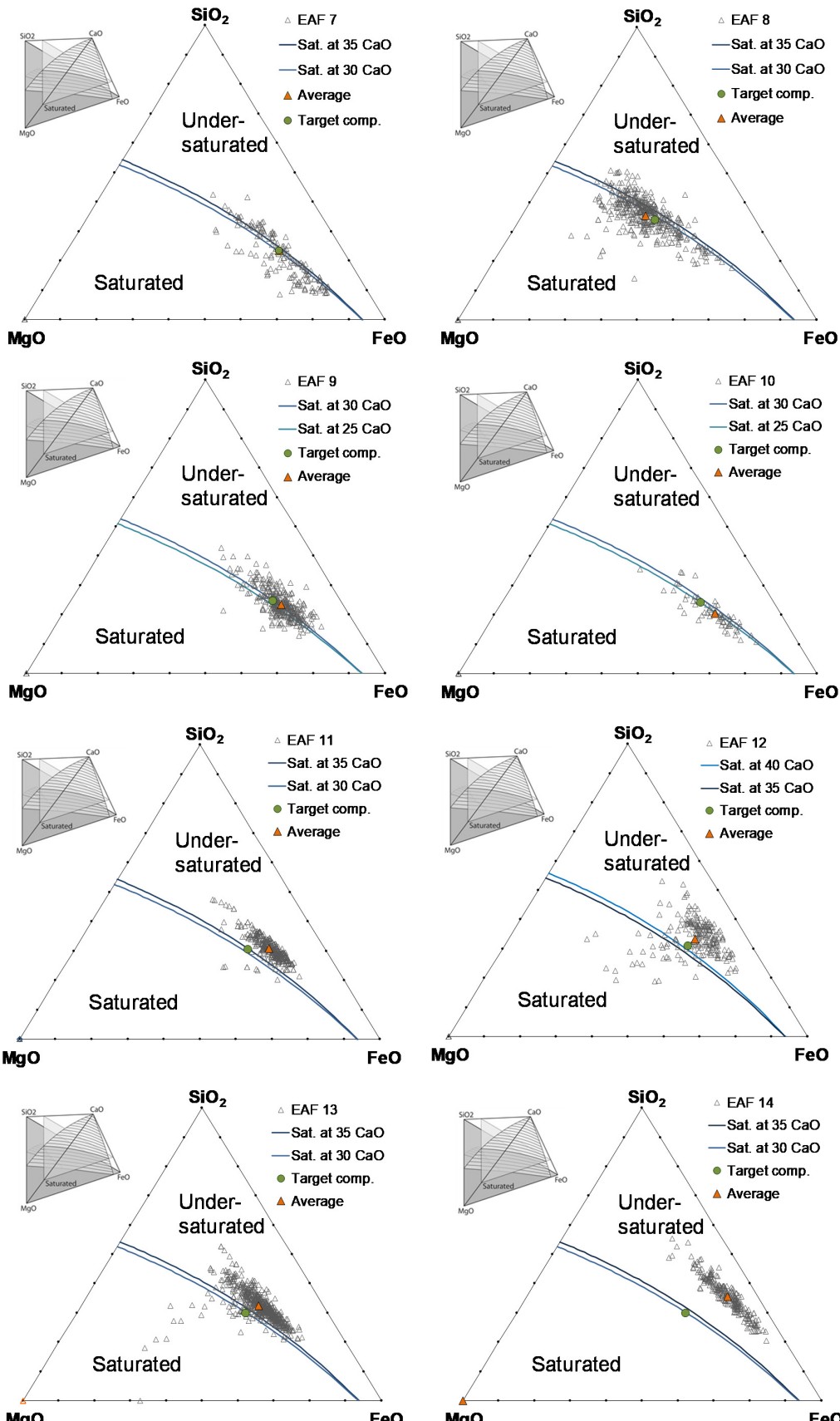

**Figure 6.** Distribution of slag analysis data from EAF heats for carbon steelmaking, in the system CaO–SiO$_2$–FeO–MgO; lines indicate MgO saturation at 30% to 40% CaO and 1600 °C.

EAF 14, however, operated systematically at MgO-undersaturated conditions, obviously with a certain control on MgO losses. As lime was applied as the only slag former, the tapped MgO content represents an approximately 1.7% MgO pick-up from the lining, however controlled, due to the rather short process time of a continuously charged EAF.

## 4. Discussion

The usefulness of the proposed MgO saturation diagram in Figure 2 was shown by applying large data sets of analyzed slag samples from industrial EAF production in Figures 4–6. Important information about the MgO saturation status of the slags and its control by production management can be derived from the distribution of the data with respect to the MgO saturation surface in the complex compositional space. In contrast, pointwise calculation of MgO saturation levels for the average composition or every single slag composition from other MgO saturation models is less informative for large datasets.

Saturation with Ca silicates occurs only at rather low FeO content, e.g., in some $Cr_2O_3$-containing stainless steelmaking slags. $Mg_2SiO_4$ forsterite occurs only below 25% CaO (shown as light gray lines in the Figure 2 tetrahedron). These conditions are usually not both realized in industrial EAF slags for carbon steelmaking. Concepts of double saturation with $C_2S$ (e.g., [4,5,12]) are of no importance for usual carbon steelmaking EAF slags. Therefore, MgO saturation was the only stability field considered here for EAF slags, bearing in mind its importance for good slag foaming and minimum refractory corrosion.

It is interesting to note that the average slag compositions for, e.g., EAFs 7, 8, 12, and 13 with low control over slag operation were still close to MgO saturation. However, MgO-undersaturated slag data indicated heats with (1) lower slag foaming and, depending on the slag volume, (2) increased radiation losses to the EAF walls and (3) increased corrosion potential of the MgO lining. The diagram presented in Figure 3 is helpful in indicating the share of inappropriate heats and providing initial information about the particular EAF control problem.

The desired MgO saturation levels (the proposed "target" composition in Figures 4–6) are defined with respect to particular EAFs and steel plant conditions, i.e., depending on the particular FeO levels and available raw materials. In the proposed representation of the MgO saturation surface, the MgO saturation conditions were investigated and defined independently from standard basicity concepts for EAF steelmaking. This might be of interest when considering EAF process slags in other oxide systems, e.g., for Cr stainless steelmaking or non-ferrous metal production.

## 5. Conclusions

Control of the slag composition in general and MgO saturation in particular plays an important role in energy-efficient EAF processing with minimum losses of iron and alloys to the tapped slag.

The application of phase equilibrium diagrams derived from internally consistent databases for steelmaking slags has been established as a common tool for interpreting plant or laboratory data with respect to saturation conditions. A diagram based on thermo-chemical calculations [32] was presented in this study, focused on MgO saturation in the system $CaO–MgO–FeO–SiO_2–5\%Al_2O_3$ and rather close to the real EAF slag compositions. Examples were shown for large datasets from 14 EAFs for carbon steelmaking, indicating that the proposed diagram is suited to interpreting data sets from industrial EAF slag samples for process optimization.

Deeper analysis for different groups of heats (e.g., those with different steel grades or different raw materials) is evidently required for improved EAF slag and operation control, e.g., [19], but not shown here due to the focus on the general applicability of the proposed MgO saturation diagram.

The diagram might be applied to monitor regularly or even on-line the particular slag operation with respect to MgO saturation and FeO control.

**Funding:** This research received no external funding. The APC was funded by the German Research Foundation (DFG) and the University of Bayreuth in the funding programme Open Access Publishing.

**Acknowledgments:** A previous version of the model was developed and published within RHI Magnesita [19]. Here we present an improved version, more profound theoretical basis, significantly more illustrations, and implications. Support for metallurgical process optimization in EAF, etc., is available from the RHI Magnesita process group.

**Conflicts of Interest:** The author declares no conflict of interest.

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
