# Peer review of "Visualization of Slag Data for Efficient Monitoring and Improvement of Steelmaking Slag Operation in Electric Arc Furnaces, with a Focus on MgO Saturation"

_metals, doi:10.3390/met11010017_

Round 1
Reviewer 1 Report
In this paper, the author has done some work about the MgO saturation in EAF slag. However, the article is not good enough, especially in the innovation and depth. So, I cannot recommend the publication of this manuscript in Metals.
In order to improve this article, some of my comments are given as following:
1. The title is too broad, but the study is just focuses on the MgO saturation in EAF slag.
2. The MgO saturation in EAF slag has been studied a lot by previous researchers, but the innovation of this paper is insufficient (no new theories, no new methods, and no new applications).
3. In this paper, the author just drew some phase diagrams using FactSage software, and read the MgO saturation from them. It is not enough in the research and analysis.
4. It is not a research article in form and content, but looks like a calculation report. For example, (1) the research method has not been described independently, (2) the content is chaos, and (3) the citation of figure and table is not normal.
5. This paper focuses on the MgO saturation in EAF slag, but it is related to the slag composition, especially SiO2 and Al2O3, which are not considered. And the saturation mechanism was not mentioned and discussed.
6. In the Introduction Section, the significance and influence of MgO saturation should be further explained: fluidity, lining erosion, current efficiency, etc. And the data in Figure 1 can be updated to 2020.
7. For the data in Table 1, the source is not explained.
8. For the second section, it seems that it can be incorporated into the first section as the research background.
9. In the third section, it is better to describe previous models in detail, and then establish an author’s model, compare and analysis it with the previous models.
10. In Figure 3~6, the distance between lines is too close to be friendly read. It is suggested to change the coordinates or enlarge part of the diagram.
11. The conclusion section is not straightforward enough.
Author Response
Dear reviewer,
thank you for you comments that helped me to improve the paper.
I thourougly revised the manuscript by reorganising and rewriting the content to a clearer structure to better meet the standards: All parts describing the state of the art are now in chapter 1 with 2 addtional paragraphs on 1.2. MgO saturation of EAF slags and 1.3. MgO saturation models... 7 references have been added. Then the derivation of the proposed diagram in chapter 2. Description of the slag data and application in chapter 3, new chapters for discussion and conclusion. Newly written text of the revised manuscript is in red for better overview.
I hope the manuscript is now better suited for publication.
Best regards, Marcus Kirschen
To your comments in detail:
1. The title is too broad, but the study is just focuses on the MgO saturation in EAF slag.
I changed the title to
Visualization of slag data for efficient monitoring and improving steelmaking slag operation in the Electric Arc Furnace, with a focus on MgO saturation.
2. The MgO saturation in EAF slag has been studied a lot by previous researchers, but the innovation of this paper is insufficient (no new theories, no new methods, and no new applications). and
3. In this paper, the author just drew some phase diagrams using FactSage software, and read the MgO saturation from them. It is not enough in the research and analysis.
The method - applying thermochemical calaculations on a predefined database - is indeed conventional. The result, however, is a diagram that was never published before by other researchers. It is very helpful for interpretation of slag data from EAF steel plants, because the derived saturation diagram is close to the real slag compositions and provides a sufficiently large composition space for teh variations that are common in industrial EAFs. All other, although precise MgO saturation models are not suitable for large datasets or visualization in the steel plant. The only suitable and widely accepted diagram for steel makers - from Pretorius 1998 - was derived by extrapolating from puiblished ternary diagrams, at the best level. However, we can do better and more precise now, e.g. with Factsage. This is shown in the paper.
4. It is not a research article in form and content, but looks like a calculation report. For example, (1) the research method has not been described independently, (2) the content is chaos, and (3) the citation of figure and table is not normal.
I reorganised the entire paper for a better structure of the content now.
5. This paper focuses on the MgO saturation in EAF slag, but it is related to the slag composition, especially SiO2 and Al2O3, which are not considered. And the saturation mechanism was not mentioned and discussed.
SiO2 and Al2O3 are considered for the saturation compositions in the system CaO-MgO-FeO-SiO2-5%Al2O3 (the abrrevated denominations of the edges in the figures 4-6 might be misleading. Figures 2 and 3 are correct)
6. In the Introduction Section, the significance and influence of MgO saturation should be further explained: fluidity, lining erosion, current efficiency, etc. And the data in Figure 1 can be updated to 2020.
All these topics are now added in chapter 1, with 7 additional references.
Figure 1 is updated wwith new data from Dec 4.
7. For the data in Table 1, the source is not explained.
Data sources, data handling and some details on the EAF plants are added now, the plants remain anonymous however, because production details are interesting but not important for the topic of this paper.
8. For the second section, it seems that it can be incorporated into the first section as the research background.
Yes. Done with 2 new paragraphs 1.2. MgO saturation of EAF slags and 1.3. MgO saturation models... The revised sections are the red paragraphs.
9. In the third section, it is better to describe previous models in detail, and then establish an author’s model, compare and analysis it with the previous models.
Yes. Now in chapter 2 with more explanations on existing models and the proposed diagram.
10. In Figure 3~6, the distance between lines is too close to be friendly read. It is suggested to change the coordinates or enlarge part of the diagram.
I removed the lines for 1650°C for better clarity in Figs 3-6 which are not necessary for the overview here.
11. The conclusion section is not straightforward enough.
I wrote the discussion section and the conclusion section newly, and focused on MgO saturation.
Reviewer 2 Report
Dear Author
Your paper "Visualization of slag data for efficient monitoring and improving steelmaking slag operation in the Electric Arc Furnace" discusses about the condition of a MgO saturation through the use of a computed pseudoternary diagram by Factsage software.
First of all, I would like to congrats with you for the clear exposed and presented paper. The manuscript has the right length and the appropriate attractiveness to become an easy-implemented instrument in any EAF steelshop.
I only highlighted some revision must be applied. You can find them directly in the attached pdf as a comment.
The only suggestion I leave here is to add a brief explanation about the data used for making figure 4 to 6.
Thus:
1) include a brief explanation about the average data collected in Table 1. In particular specify the number of slag used to compute such average.
2) add the method you follow to group the data into your 14 categories
3) add a brief explanation about the data used in figure 4 to 6, maybe specifying that the available data differ in number from a group to another because they are related to different plants
4) add a brief information about the origin of the steelshops supplier for slag data
best regards

Author Response
Dear reviewer,
thank you for your comments. Very helpful to improve the paper.
I revised the paper thouroughly due to the comments from another reviewer and I added the information that you asked for:
1) include a brief explanation about the average data collected in Table 1. In particular specify the number of slag used to compute such average. and
4) add a brief information about the origin of the steelshops supplier for slag data
I added a paragraph before table 1 and footnotes in table 1 to provide more information about the steel shops, only general information to maintain confidentiality:
Average slag compositions of 14 industrial EAFs for low alloyed carbon steelmaking are given in Table 2. Product portfolio covered rebar and construction steel grades (8 EAFs) to special steel grades (2 EAFs). Applied raw materials are steel scrap (10 EAFs) and blends of steel scrap with DRI (4 EAFs). Sizes of the EAFs ranged from 60 t to > 200 t tap weight, located in 9 countries worldwide. Slag samples have been taken from the EAF shortly before tapping and analyzed at the steelplant laboratory. Only mislabeled slag data, e.g. from transport ladles (i.e. FeO < 10% and CaO > 45%) or from raw materials (e.g. lime, DRI) have been excluded from the data sets.
2) add the method you follow to group the data into your 14 categories
3) add a brief explanation about the data used in figure 4 to 6, maybe specifying that the available data differ in number from a group to another because they are related to different plants
Numbers are added in table 1 now.
The examples in Figures 4-6 represent large datasets about 14 EAFs operations, that are remarkabyl different from each other, although 12 EAFs are rebars/constrction steel grades. The focus here is to prove the applicability of the proposed diagram. Of course, in each individual study we had a closer look on the data and grouped either for time evolution or different steel grades, for adaoted input of slag formers for example. This was neglected in the manuscript. We just applied all data to the figures to see the coincidence with the calculated lines.
Best regards, Marcus Kirschen
Reviewer 3 Report
This is an interesting paper to control slag compositions in the electric arc furnace for low CO2 steel making. The author uses the FactSage database to understand the MgO saturation composition and compared with various EAF slag data. The analysis is useful to minimize try and error approach especially for monitoring MgO concentration. This paper is worthy of publication in the Metals.
Author Response
Dear reviewer,
thank you for your supporting comments.
Best regards, Marcus Kirschen
Reviewer 4 Report
In Figure 4-6 it is not clear why in the legend of the diagrams the MgO saturation lines are mentioned twice with the same text.
In the title of these figures there is a typing error: "form" instead of "from"
Author Response
Dear reviewer,
thank you for your supporting comments.
Figures 3-6 have been revised for better readability and clarity. The double lines represented 1600°C and 1650°C which was confusing. 1650°C is now removed, and the figures are better to read. The typo was also corrected.
Best regards, Marcus Kirschen
Round 2
Reviewer 1 Report
It can be accepted.
Reviewer 2 Report
Dear Author,
Thank you for providing a revised version of your paper
I have not more comments about. The paper is suitable for publication at the current state
Best regards